# Modeling of Triphenyl Phosphate Surfactant Enhanced Drying of Polystyrene/*p*-Xylene Coatings Using Artificial Neural Network

Devyani Thapliyal [1], Rahul Shrivastava [2,*], George D. Verros [3], Sarojini Verma [1], Raj Kumar Arya [1,*], Pramita Sen [4], Shiv Charan Prajapati [5], Chahat [6] and Ajay Gupta [7]

1 Department of Chemical Engineering, Dr. B.R. Ambedkar National Institute of Technology, Jalandhar 144011, India; devyanithapliyal5@gmail.com (D.T.); vermasarojini488@gmail.com (S.V.)
2 Department of Chemical Engineering, Jaypee University of Engineering & Technology, Guna 473226, India
3 Laboratory of Chemistry and Technology of Polymers and Dyes, Department of Chemistry, Aristotle University of Thessaloniki, 54124 Thessaloniki, Greece; gdverros@gmail.com
4 Department of Chemical Engineering, Heritage Institute of Technology, Kolkata 700107, India; pramita.sen@gmail.com
5 Department of Paint Technology, Government Polytechnic Bindki, Fatehpur 212635, India; scprajapati@yahoo.com
6 Computing Studies & Information Systems, Douglas College, New Westminster, BC V3L 5B2, Canada; lutherchahat@gmail.com
7 Department of Industrial and Production Engineering, Dr. B.R. Ambedkar National Institute of Technology, Jalandhar 144011, India
* Correspondence: rahul.shrivastava@juet.ac.in (R.S.); rajaryache@gmail.com (R.K.A.)

**Abstract:** The drying process of polymeric coatings, particularly in the presence of surfactants, poses a complex challenge due to its intricate dynamics involving simultaneous heat and mass transfer. This study addresses the inherent complexity by employing Artificial Neural Networks (ANNs) to model the surfactant-enhanced drying of poly(styrene)-p-xylene coatings. A substantial dataset of 16,258 experimentally obtained samples forms the basis for training the ANN model, showcasing the suitability of this approach when ample training data is available. The chosen single-layer feed-forward network with backpropagation adeptly captures the non-linear relationships within the drying data, providing a predictive tool with exceptional accuracy. Our results demonstrate that the developed ANN model achieves a precision level exceeding 99% in predicting coating weight loss for specified input values of time, surfactant amount, and initial coating thickness. The model's robust generalization capability eliminates the need for additional experiments, offering reliable predictions for both familiar and novel conditions. Comparative analysis reveals the superiority of the ANN over the regression tree, emphasizing its efficacy in handling the intricate dynamics of polymeric coating drying processes. In conclusion, this study contributes a valuable tool for optimizing polymeric coating processes, reducing production defects, and enhancing overall manufacturing quality and cost-effectiveness.

**Keywords:** ANN modeling; surfactant enhanced drying; thin films; poly(styrene); triphenyl phosphate

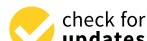



## 1. Introduction

Chemical substances known as surfactants are used to reduce the interfacial tension between a solution and another phase by forming micelles. Ionic and non-ionic surfactants are the primary kinds of surfactants [1]. Different categories of surfactants can only be used if they are compatible with the coating solution. Because of their interactions, drying rate, film formation, wetting performances, and other uses, polymer-surfactant systems are currently of significant research interest. The hydrophobicity of polymer influences the interaction between polymers and surfactants. The aggregate number of polymer bounds is independent of the bound surfactant concentration [2].

In polymeric films, even a modest amount of surfactant causes a noticeable alteration. Surfactant–polymer interaction requires a specific surfactant concentration. If the value is less than this, aggregation causes the addition of surfactant to increase the viscosity of the solution. Above this point, increasing the amount of surfactant results in steric hindrance in the association, which results in a low aggregation number, and the viscosity drops below its ideal level [3]. By reducing surface tension close to the edges, surfactant produces a surface tension gradient between the edges and the center of the film. Because of the Marangoni force, a flow is produced that moves the solute back to the film's center. This makes the polymeric coatings flat [4]. Surfactants significantly lower the glass transition temperaturereduced. Amphiphilic compounds, which are typically used as plasticizers, are known for this behavior [5].

The drying process of polymeric films is crucial for manufacturing products like photographic films, adhesive tapes, and coatings. Controlled evaporation of solvents is the key, with temperature, humidity, and drying equipment playing vital roles. The process influences film thickness, uniformity, and microstructure, impacting the final product's quality. Techniques such as spectroscopy ensure quality control during drying. Overall, understanding and optimizing the drying mechanism is essential for achieving consistent product quality, energy efficiency, and cost-effective manufacturing. Our previous research groups studied the effect of various surfactants, including anionic, non-ionic, and fluoro-surfactants, to minimize the residual solvent and enhance the drying rate of organic and water-based coatings [6–9]. Triphenyl phosphate (TPP) was used as a plasticizer in one of the studies to modify the drying behavior of the poly(styrene)-*p*-xylene system in order to minimize the residual solvent [8]. The amount of plasticizer present significantly impacts the amount of residual solvent left after drying. A machine learning technique based on a regression tree model was also used to develop a drying model for a poly(styrene)-*p*-xylene system enhanced by triphenyl phosphate as the surfactant [9]. The predictions of the developed model based on regression trees were obtained by MATLAB software (R2014a). It was robust and acceptable to be employed with the system regardless of composition or thickness. With regard to estimating weight loss for a specific time, TPP, and initial thickness values, the model derived results. Within 1% of the experimental data, the model predictions were accurate.

Artificial Neural Networks (ANNs) have recently gained popularity and have proven to be a useful model for classification, clustering, pattern recognition, and prediction in a variety of fields [10]. ANNs, a machine learning model, are now comparable with traditional statistical and regression models in terms of usefulness [11]. The ANN is discovered to be a very new and practical paradigm applied to machine learning and problem-solving. ANN is comparable to how the human brain's nervous system operates [12]. A neural network is a computational model in which many nodes (or neurons) are interconnected [13]—a distinct output function known as the activation function is represented by each node. Every connection between two nodes represents a weight for the signal traveling across the connection, comparable to the Artificial Neural Network's memory. The output will change depending on the network's connectivity, weight value, and incentive function [14,15].

The immense potential of Artificial Neural Networks (ANNs) lies in their high-speed processing through massively parallel implementation, prompting increased research in this field. ANNs, known for their exceptional attributes such as self-learning, adaptivity, fault tolerance, and non-linearity, are primarily utilized for universal function approximation in numerical paradigms. In direct comparison, statistical models offer few advantages over ANNs. The ANN approach does not necessitate a deep understanding of underlying phenomenological mechanisms, established mathematical equations, explicit expressions, or input–output interactions. Unlike statistical models, there are no presumptions about the distribution or characteristics of the data in ANN models. ANNs serve as highly adaptable data reduction models, encompassing discriminant and non-linear regression models [16,17].

In surface coatings, ANNs have a broad range of applications, including coating thickness prediction, coating hardness prediction, microstructure analysis, roughness, particle characteristics, etc. They can also predict the hysteresis effect in sputtering processes and coating oxidation behavior. By rejecting undesired sounds and making up for the modified variables, ANN is a key player in the process modeling of surface coatings [18]. In order to assess the predictions regarding the wear amounts of surface coatings applied by the welding melting method, Ulas et al. [19] developed models using four different machine learning algorithms (ANN, extreme learning machine, kernel-based extreme learning machine, and weighted extreme learning machine). The study showed that, to differing degrees, machine learning systems can accurately anticipate the wear loss quantities of various coated surfaces. Barletta et al. [20] used a multi-layered perceptron (MLP) neural network to model the electrostatic fluidized bed coating process to predict the thickness trends with respect to time, voltage, and airflow.

The extensive use of Artificial Neural Networks (ANNs) in drying applications has been historically driven by their advantageous properties. In various drying technology methods such as batch convective thin-layer drying, fluidized bed drying, osmotic dehydration, osmotic-convective drying, spray drying, freeze drying, rotary drying, and spout bed drying, ANNs have been employed to model, predict, and optimize heat and mass transfer. They have also played a crucial role in studying thermodynamic parameters and physiochemical properties of dried products [21–26]. However, despite this widespread application, there has been a notable gap in the literature concerning the drying of binary polymeric solutions and surfactant-enhanced drying of polymeric solutions using the ANN approach. This indicates an untapped potential for leveraging ANNs in understanding and optimizing these specific drying processes, marking an opportunity for future research endeavors in this domain.

The drying of polymeric solution coatings is a complex phenomenon involving simultaneous heat and mass transfer, further complicated by intricate thermodynamics spanning from dilute solutions to highly concentrated, i.e., glassy state. This process encompasses the diffusion of solvent(s) within the coating, followed by evaporation from the surfaces. The diffusion in binary polymeric coatings is well-characterized using the Vrentas and Duda free volume theory [27,28]. A typical binary diffusion model involves 13 free volume theory parameters, with 9 being pure component properties and the remaining 4 estimated from drying experiments through regression.

To minimize production defects in coatings, it is crucial to reduce residual solvent contents. This can be achieved by optimizing drying conditions, incorporating additives to enhance drying [6,8], or a combination of both. The addition of any material to the coating solution significantly affects the solution's thermodynamics, diffusion, and available free volume. Currently, no diffusion model is available to predict the drying behavior for surfactant-enhanced drying.

The drying process of polymeric coatings, inherently intricate due to the multitude of components involved, including polymers, solvents, binders, pigments, leveling agents, and drying enhancers like surfactants, has been a subject of conceptual modeling. The gradual nature of polymeric coating drying often results in a significant accumulation of residual solvents during the later stages. This surplus amount of solvent, if not effectively managed, can lead to coating defects and substantial solvent losses. To address this challenge, surfactants are introduced into the coatings to expedite the drying process, facilitating the removal of a maximum amount of solvent within a shorter timeframe.

Remarkably, despite the critical role of surfactants in enhancing drying rates and minimizing solvent retention, a notable gap exists in the form of a lack of a comprehensive mathematical model. The current body of knowledge lacks a predictive tool that can accurately forecast the intricate dynamics of drying and the specific impact of surfactants on the drying rate in this complex system.

This work aims to bridge this gap by developing a sophisticated mathematical model tailored to this specific context. Leveraging existing drying data, this endeavor seeks to

provide a comprehensive and predictive understanding of the drying process in polymeric coatings, shedding light on the nuanced influence of surfactants on drying rates. This novel approach represents a significant advancement in the field, offering valuable insights and practical applications for optimizing polymeric coating processes.

In this work, we report the application of Artificial Neural Networks (ANN) to model surfactant-enhanced drying in a binary solution. The system employed uses triphenyl phosphate as a surfactant to study the drying behavior of poly(styrene)-*p*-xylene coatings, with experimental data points sourced from our earlier work [8]. In today's context, a major task is to develop simple and robust methods for the online control of dryers, manipulating control variables such as temperature or airflow to achieve the best quality product at a minimum cost. This work aims to test these algorithms on a laboratory scale.

## 2. Materials and Methods

Before the emergence of machine learning techniques, researchers have used design of experiment (DOE) methods to determine the relationship between various factors affecting the process and its outputs; in other words, causal analysis is used to identify cause-and-effect relationships. There are multiple approaches to DOE, like Taguchi and response surface methodology (RSM). For example, the precipitation of barium sulfate salt was investigated through dynamic tube-blocking tests using RSM [29].

After the emergence of artificial intelligence and machine learning techniques (MLTs), more competitive methods have been developed, which can be adapted to complement or replace traditional RSM methods. In contemporary scientific and engineering research, MLTs and artificial intelligence have gained widespread popularity for modeling complex systems. Various MLTs, including Artificial Neural Networks (ANN), support vector regression machines (SVM), and regression trees (RT), among others, are available for this purpose. Numerous instances in the literature showcase the successful application of these techniques across diverse scientific and engineering domains.

For instance, an ANN was employed to develop a model predicting the physical and chemical properties of aqueous extracts from nine medicinal plants. This model considered dynamic experiments based on extraction conditions (time and temperature) and plant species [30]. Another study utilized an ANN and multiple regression-based model to predict organic potato yield based on the tillage system and soil properties [31]. Accurate solar irradiance forecasts, crucial for solar energy system operators, were achieved using an ANN to capture the non-linear relationship between solar irradiance and atmospheric variables [32].

In assessing groundwater suitability for irrigation, various methods, including indexical approaches, statistical computing, graphical plotting, and machine learning algorithms, were compared, with the ANN outperforming other techniques [33]. A support vector machine-based model was employed to assess the behavior of waste tire rubberized concrete [34]. Different approaches, such as ANN and regression tree simulations, were used to investigate the I-V characteristic of an ion-sensitive field-effect transistor based on graphene [35].

A regression tree-based model successfully predicted these quantities to overcome the laborious process of experimentally determining a compound's water solubility and Setschenow coefficient [36]. Another model based on a regression tree was developed to assess the importance of plant, soil, and management factors affecting potential milk production on organic pastures [37].

These MLTs, functioning on the black box principle with historical/experimental data, eliminate the need for explicit system knowledge. When mathematical modeling for complex processes is challenging, conducting numerous experiments to generate sufficient input–output data enables the development of machine learning models. These models exhibit good generalization, accurately predicting output for given inputs, and are applicable to unseen data.

This study chose an ANN model due to its popularity and high prediction accuracy to forecast weight loss in coatings. This prediction was based on input variables such as time,

the amount of triphenyl phosphate (TPP) used as a surfactant, and the initial thickness of the coating. In a previous study, a regression tree was used for the same system [9]. While an ANN has more parameters than a regression tree, its flexibility allows for capturing complex non-linear relationships. However, drawbacks include the empirical nature of model creation, increased computational load, and the risk of overfitting. In contrast, the regression tree is a straightforward method, building rules based on features in the data.

*Model Development*

A brief description of ANN has been provided in this section. For a detailed description, the reader can refer to [38].

A neural network is a group of interconnected neurons that are taught from their environment to take in linear and non-linear trends in complicated data to produce reliable results for unseen situations containing even noisy and incomplete information [39].

In an Artificial Neural Network (ANN), there are three layers: the input, hidden, and output. There may be more than one hidden layer. Each layer contains multiple neurons. Each neuron in a layer has an activation function. The activation function is the basis of non-linearity in the relationship.

A neuron emits one signal, known as output, after ingesting many signals, collectively called inputs. Figure 1 depicts the McCulloch and Pitts (MCP) simple neuron.

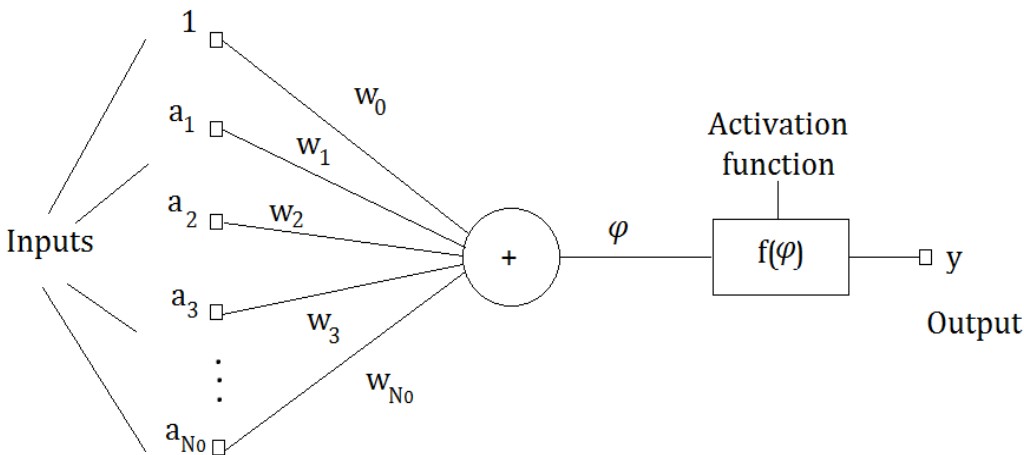

**Figure 1.** A simple MCP neuron.

Each input is assigned a weight. Each input's impact on the decision-making process is influenced by its weight. Weighted input is created by multiplying the input by its weight. The MCP neuron can adjust to a given circumstance by altering its weights. Several methods are available for that aim, including the Delta rule and back error propagation. Thus, a neuron comprises two main components: weight and activation function. The weights decide the input vector's strengthdetermined. The output of the node is excited by a +ve weight and inhibited by a −ve weight.

$$\varphi = x^T w = x_1 w_1 + x_2 w_2 + x_3 w_3 + \ldots + x_n w_n = \sum_{i=1}^{n} x_i w_i \tag{1}$$

In order to create the final output ($y$), a mathematical procedure involving an activation function ($f$) is applied to the signal output.

$$y = f(\varphi) = f\left\{\sum_{i=1}^{n} x_i w_i\right\} \tag{2}$$

Linear (or ramp), hyperbolic tangent, and sigmoid activation functions are the most frequently employedcommonly.

ANNs contain many simple, highly interconnected neurons. There are two different kinds of networks: feed-forward networks and feedback networks. Signals move from

the input via hidden layers to the output in a feed-forward network, while in a feedback network, signals are fed back from the output to hidden layers. Only feed-forward neural networks have been used in this study due to their simplicity.

Feed-forward ANNs only allow one direction of signal flow: from input to output. No feedback loops exist at all. Figure 2 depicts a feed-forward network.

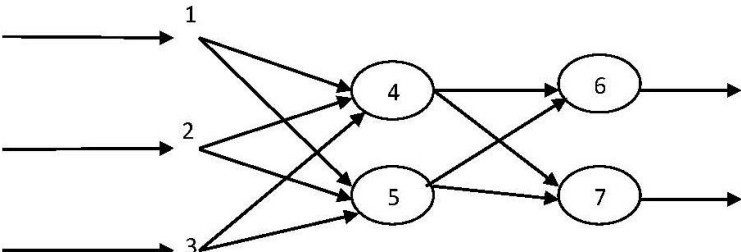

**Figure 2.** An example of feed-forward network with hidden layer.

The nodes are images of neurons. Arrows show the connections between nodesindicated. The input nodes of the network are input variables $(x_1, x_2, \ldots, x_n)$, which only transmit values to processing nodes without performing any computations, while the output variables $(y_1, \ldots, y_n)$ are the output nodes. Nodes can be placed in layers, such as the input layer, which comprises nodes 1, 2, and 3; the hidden layer, which includes nodes 4 and 5; and the output layer, which consists of nodes 6 and 7. Several hidden layers could be present in a neural network.

A set of weights ($w_{ij}$) are present in the *jth* node of a network. As an illustration, node 4 has weights $w_{14}$, $w_{24}$, and $w_{34}$. A network is considered complete if the set of all weights ($w_{ij}$), the topology (or graph), and the activation function of each node are all known.

In this investigation, the chosen neural network architecture is the most typical type—an Artificial Neural Network (ANN) with one hidden layer. This configuration is widely used and comprises an input, hidden, and output layer. The hidden layer has a fixed number of nodes, but determining the optimum number of hidden nodes is a crucial aspect of network design. Additionally, more complex configurations, such as multilayer networks with multiple hidden layers, are plausible. However, establishing the ideal number of hidden layers and neurons within each layer lacks a universally accepted method.

The connections' strengths within the network, represented by weights, are determined through various methods. One prevalent technique is supervised learning, a process wherein the neural network is trained by exposing it to teaching patterns, allowing it to adjust its weights based on a specified learning rule. In this method, the network receives a set of input data and a corresponding set of desired output data (target values). The network's output is then compared to the expected output. If any discrepancy arises, the weights are adjusted by the learning rule until the network's output aligns with the desired output.

In contrast, unsupervised learning is another training approach where the network adapts to structural elements in input patterns without explicit target outputs. The network learns autonomously through this method. However, for this investigation, the chosen approach is supervised learning.

Various strategies can be employed to minimize the discrepancy between the network's output and the desired output (target value) in supervised learning. These strategies include the sum of squared errors, least absolute deviation, asymmetric least squares, percentage differences, and least fourth powers. Among these, the sum of squared errors is the most commonly used metric for training neural networks, serving as a measure to guide the adjustment of weights during the learning process.

The backpropagation technique is a crucial mechanism employed to reduce prediction errors within a neural network. This iterative process involves the successive adjustment of weights based on the computed errors, ultimately refining the network's performance. The technique unfolds through the network's layers, starting from the first hidden layer,

which receives inputs, calculates outputs, and forwards them to subsequent layers until reaching the output layer.

At each output layer unit, activation is determined by summing the weighted inputs from all preceding layers. The difference between the desired output and the actual network output is termed as the error. This error is then used to adjust the weight matrix, and the process iterates until the error is minimized.

A key parameter in this process is the number of epochs, indicating how often the weights have been updated during the network training. The number of epochs plays a pivotal role in influencing the model's performance. As the number of epochs increases, the data may transition from underfitting to optimal fitting and potentially to overfitting. However, the number of epochs alone is not a decisive factor; more critical are the validation and training errors. Training should continue as long as these errors consistently decrease.

Another essential hyperparameter is the learning rate, representing the step size during the minimization of a loss function. This parameter metaphorically signifies how rapidly a machine learning model "learns". The learning rate determines the extent to which new information supersedes previous knowledge. Selecting an appropriate learning rate involves a trade-off between convergence rate and overshooting. A high learning rate may lead to skipping minima, while a low learning rate might result in slow convergence or getting stuck in an unfavorable local minimum. Striking the right balance is crucial for effective and efficient neural network training.

Assume a training set $\{(x_1, t_1)...(x_r, t_r)\}$ that consists of r-ordered pairs of $n \times m$ dimensional vectors, referred to as input and output patterns. A continuous and differentiable activation function is required at each network node. Random initialization is used for the weights. For the given input pattern $(x_i)$ from the training set, the network will produce output $y_i$, which will be compared with the target value $(t_i)$. The final objective is to use a learning algorithm to reduce the disparity as much as possible between $y_i$ and $t_i$ for $i = 1...r$. To be more precise, we want to minimize the network's error function. ANNs can be trained by using any one of the following learning algorithms.

- Gradient descent
- Levenberg–Marquardt algorithm
- Newton method
- Quasi–Newton method
- Conjugate gradient

In this study, the widely used structure of the neural network, i.e., the backpropagation feed-forward network with one hidden layer, has been used.

## 3. Results and Discussion

ANN-based methods are widely used. Numerous trustworthy open-source libraries, including Python and R software, are readily available. However, MATLAB® (The Math-Works, Inc., Natick, MA, USA) was used in this work to generate the results.

By entering the command *nnstart*, one can launch the Neural Network GUI (guide user interface) in MATLAB. A window with launch buttons for the following apps: Neural Net Fitting, Neural Net Pattern Recognition, Neural Net Clustering, and Neural Net Time Series are displayed. Links to lists of data sets, examples, and other helpful resources are also provided. The Neural Net Fitting software can be chosen for the regression model. Once the input and output data have been loaded, you can choose the percentage of the data for training, validation, and testing, the number of neurons in the hidden layer, and the training algorithm. Then, one can begin training. If not getting the results someone is looking for, you can vary the number of neurons in the hidden layer.

In this study, all 16,258 experimentally obtained samples underwent processing in the Neural Net Fitting software. This software, adapts at handling large datasets, systematically divides the data into three subsets: 70% (11,380 samples) for training, 15% (2439 samples) for validation, and another 15% (2439 samples) for testing. The decision to adopt a 70/15/15 split instead of the more common 80/20 rule was driven by the specific

characteristics of our dataset and the research objectives. While the 80/20 rule is often employed, especially with limited data, our larger dataset allowed for a nuanced approach that prioritized a substantial training set (70%) while allocating reasonable portions for validation and testing. This balance aimed to ensure effective model training and robust evaluation of its performance.

The software utilizes a sophisticated algorithm that systematically and randomly selects data for each phase—training, validation, and testing. This algorithm ensures a comprehensive representation of the dynamics inherent in the underlying process for each category. By introducing randomness into the selection process, the algorithm prevents biases that might arise from specific temporal or spatial patterns in the data. This approach enhances the model's capacity to learn underlying patterns within the training data and validates its performance on unseen data through a well-curated validation set. The testing set, derived with the same strategic randomness, serves as a rigorous benchmark for assessing the model's overall predictive accuracies.

In terms of the neural network architecture, a single hidden layer backpropagation neural network was employed in this study. The architecture featured sigmoid hidden neurons and linear output neurons. A learning rate of 0.01 was chosen to provide the optimal trade-off between convergence and overshooting during the training process. The careful consideration of data splitting and neural network parameters reflects a deliberate and systematic approach to ensure the reliability and generalizability of the developed model.

Table 1 shows the performance of ANN with variations in the number of neurons in the hidden layer.

**Table 1.** Performance of ANN with variation in number of neurons in hidden layer.

| Number of Neurons | CPU/Training Time (s) | MSE | $R^2$ Value |
|---|---|---|---|
| 5 | 54 | $2.55 \times 10^{-06}$ | 1 |
| 10 | 187 | $5.61 \times 10^{-08}$ | 1 |
| 15 | 211 | $2.37 \times 10^{-08}$ | 1 |
| 20 | 255 | $1.09 \times 10^{-08}$ | 1 |
| 25 | 81 | $2.68 \times 10^{-08}$ | 1 |
| 30 | 356 | $7.19 \times 10^{-09}$ | 1 |
| 35 | 373 | $5.50 \times 10^{-11}$ | 1 |
| 40 | 273 | $7.32 \times 10^{-11}$ | 1 |
| 45 | 101 | $1.19 \times 10^{-08}$ | 1 |
| 50 | 326 | $5.50 \times 10^{-09}$ | 1 |
| 100 | 467 | $2.05 \times 10^{-09}$ | 1 |

In this table, the second column shows the time required to train the neural network. On increasing the number of neurons, the number of weights increases, in general, which leads to an increase in training time. However, in some cases, fewer iterations are required to find the optimum values of the weights, which causes a reduction in training time. As we can see in Table 1, a network with 25 neurons took less time to be trained than a network with 10, 15, and 20 neurons. It is advised to use the fewest possible hidden neurons to complete the task. Complexity will increase if we use more hidden neurons than we need. Just with 10 neurons in the hidden layer, we could achieve the best performance in terms of complexity, training time, and R-value. The neural network's training was performed using the Levenberg–Marquardt technique. R-value and Mean Squared Error (MSE) were used as performance criteria (see Table 1). The average squared difference between outputs and targets is known as the MSE. A zero MSE value indicates no error. Regression R values quantify the relationship between outputs and targets. An R-value of "1" indicates a strong association, while "0" indicates a random relationship. The following is the formula for MSE:

$$MSE = \frac{1}{n} \sum_{i=1}^{n} (y_i - t_i)^2 \tag{3}$$

where $y_i$ is the predicted output (predicted by the model), $t_i$ is the actual output (target) for the specified sample, and n is the total number of samples. The following formula can be used to determine the $R^2$ value after computing MSE and variance in output data (y).

$$R^2 = \left[ 1 - \frac{MSE}{variance(y)} \right] \tag{4}$$

Figure 3 presents the validation performance curve depicting the Mean Squared Error (MSE) as the chosen loss function for the neural network in this study. The visualization of both training and validation losses on the graph is a common practice to assess the model's performance at selected parameter values. High losses in both categories suggest underfitting, while low training loss and high validation loss indicate overfitting. Optimal model fit is achieved when training and validation losses stabilize at a specific point. This study obtained the best fit at the 139th epoch, where the MSE reached an exceptionally low value of $7.3258 \times 10^{-8}$.

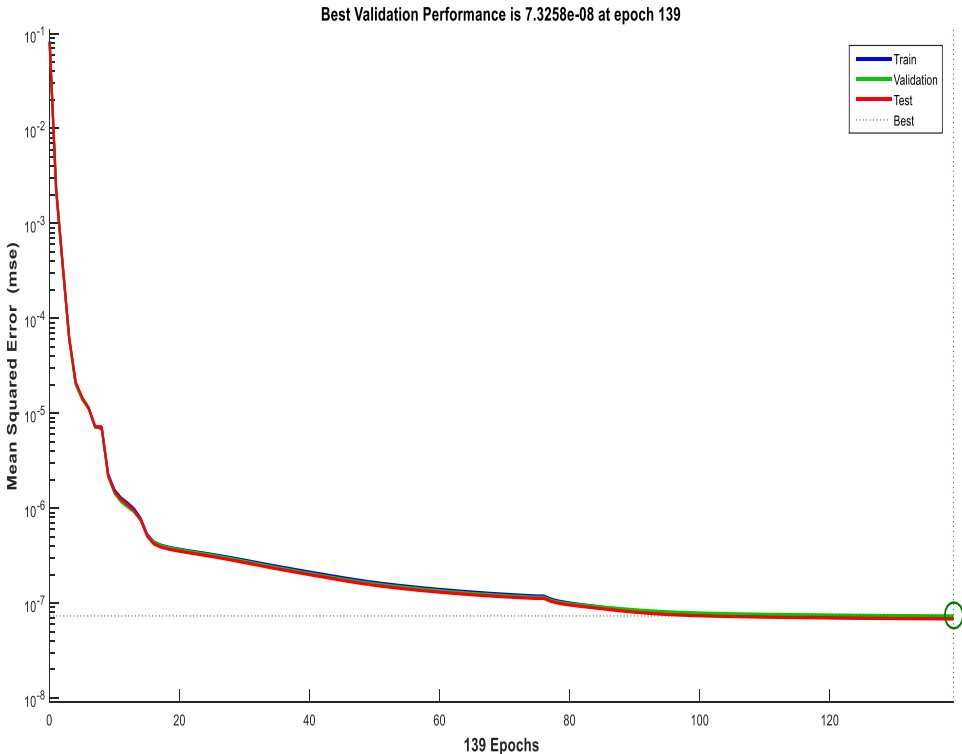

**Figure 3.** Validation performance curve.

Figure 4 showcases the regression curves generated by the Artificial Neural Network (ANN) for the training, validation, test sets, and the overall dataset. Following the training phase with 11,380 samples and validation with 2439 samples, the model's predictive efficacy was evaluated using a separate set of testing data comprising 2439 samples. Notably, these testing samples were not utilized in the ANN model's training or validation stages.

These regression curves visually represent the ANN's performance across different datasets, offering insights into how well the model generalizes to unseen data. The distinct curves for training, validation, and testing sets, along with the overall dataset, contribute to a comprehensive understanding of the model's predictive capabilities. The visual assessment aids in gauging the ANN's ability to capture underlying patterns and make accurate predictions across diverse scenarios, reinforcing the reliability of the developed model.

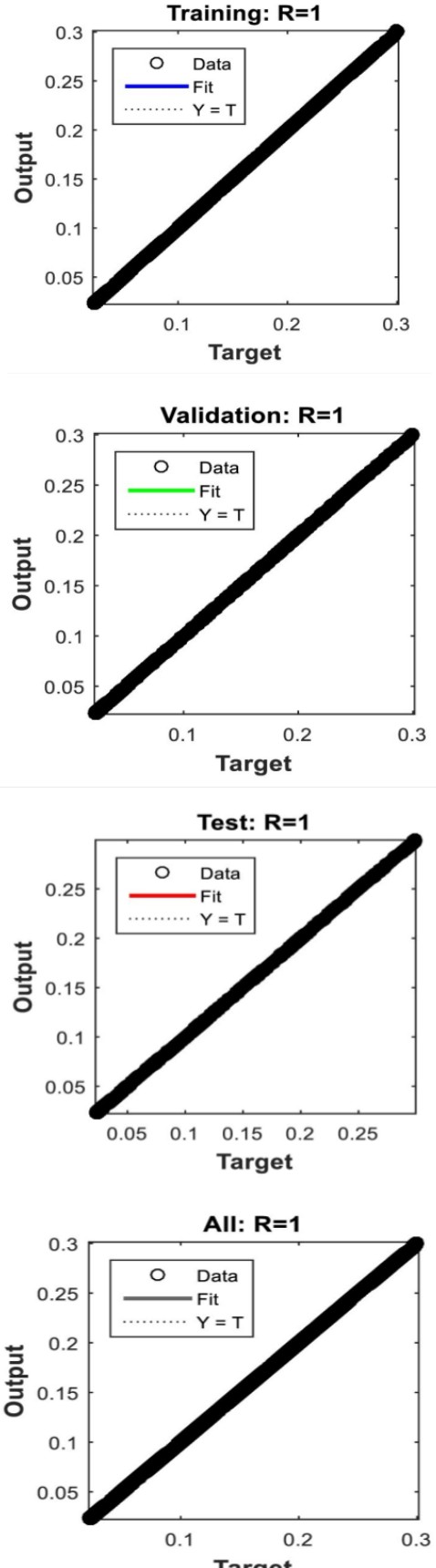

**Figure 4.** ANN regression curves with training, validation, testing sets, and overall dataset.

In Figure 4, the ANN regression curves offer a detailed insight into the predictive prowess of the model across different datasets—training, validation, testing sets, and the overall dataset. These curves juxtapose the actual output (Target, T) against the model-predicted output (Output, Y), with the *x*-axis representing target values and the *y*-axis representing outputs.

The curves serve as a critical tool for assessing the model's accuracy, with a perfect prediction represented by a straight line passing through the origin with a slope of 1. In Figure 4, the dotted line illustrates this ideal scenario where Y equals T. The 'o' markers depict the model-predicted output against the actual target value, while the solid line represents the fit to the data. Significantly, all four curves—training, validation, testing, and the overall dataset—exhibit straight lines passing through the origin with a slope of 1. This consistency indicates that the outputs predicted by the network closely match the target values. Each 'o' marker precisely aligns with the dotted line, underscoring the model's remarkable accuracy, surpassing a 99.9% precision rate in each curve.

Transitioning to Figure 5, the error histogram provides a granular view of prediction errors for all training, validation, and testing samples. The *x*-axis represents the error, indicating the disparity between target and output values, while the *y*-axis denotes the instances or samples. A discernible pattern emerges from the histogram, illustrating that the prediction error for each sample is exceptionally low, hovering around zero. This implies that the variance between target and output values is either less than zero or very close to zero for every sample.

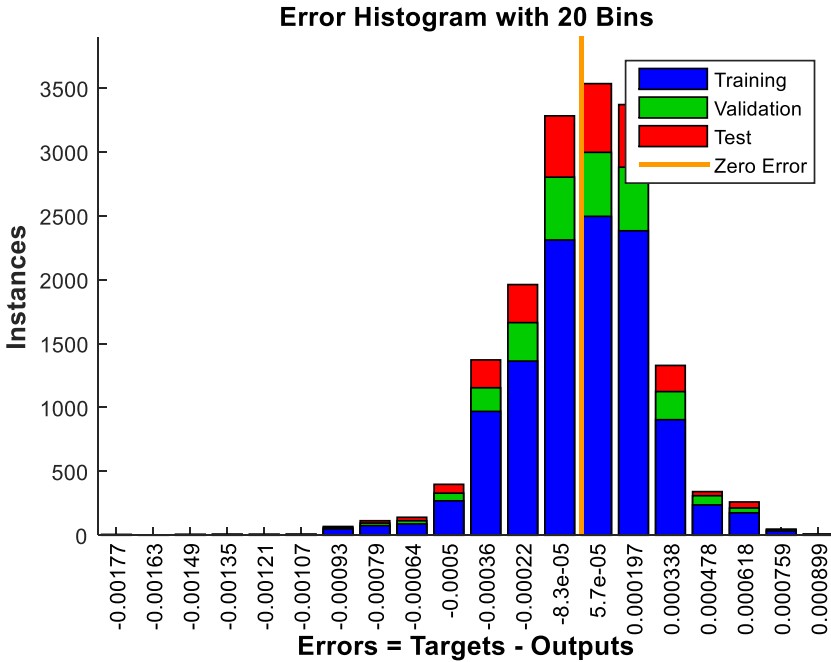

**Figure 5.** Error histogram for all the samples/instances.

In summary, the regression curves and the error histogram affirm the robustness of the developed ANN model. The consistent achievement of near-perfect predictions across diverse datasets underscores the model's reliability and accuracy in capturing the underlying patterns of the studied phenomenon. The combination of these analyses provides a comprehensive understanding of the model's performance, instilling confidence in its practical applicability for optimizing polymeric coating processes by ensuring accurate predictions of drying rates under various conditions.

Table 2 shows the results of a few samples because it is impossible to show the results in tabular form for all the 16,258 samples, but Figure 5 indicates the results for all the samples.

**Table 2.** Target and predicted values for given inputs.

| X1 × 10⁴ (Time, s) | X2 × 10⁴ (Amount of Surfactant, gm) | X3 × 10⁴ (Initial Coating Thickness, μm) | T, (Experimental Weight of Coating, gm) | Y, (Model Predicted Weight of Coating, gm) | \|% *error*\| |
|---|---|---|---|---|---|
| 0.1861 | 0.00015 | 0.2005 | 0.26 | 0.26 | 0 |
| 0.2151 | 0.0002 | 0.2009 | 0.26 | 0.26 | 0 |
| 0.1905 | 0 | 0.2021 | 0.26 | 0.26 | 0 |
| 0.1856 | 0.00015 | 0.2005 | 0.26 | 0.26 | 0 |
| 0.2146 | 0.0002 | 0.2009 | 0.26 | 0.26 | 0 |
| 0.19 | 0 | 0.2021 | 0.26 | 0.26 | 0 |
| 0.1851 | 0.00015 | 0.2005 | 0.26 | 0.26 | 0 |
| 0.2141 | 0.0002 | 0.2009 | 0.26 | 0.26 | 0 |
| 0.1895 | 0 | 0.2021 | 0.26 | 0.26 | 0 |
| 0.2136 | 0.0002 | 0.2009 | 0.26 | 0.26 | 0 |
| 0.1846 | 0.00015 | 0.2005 | 0.26 | 0.26 | 0 |
| 0.189 | 0 | 0.2021 | 0.26 | 0.26 | 0 |
| 0.2131 | 0.0002 | 0.2009 | 0.26 | 0.26 | 0 |
| 0.1841 | 0.00015 | 0.2005 | 0.26 | 0.26 | 0 |
| 0.1885 | 0 | 0.2021 | 0.26 | 0.26 | 0 |
| 0.2126 | 0.0002 | 0.2009 | 0.26 | 0.26 | 0 |
| 0.1836 | 0.00015 | 0.2005 | 0.26 | 0.26 | 0 |
| 0.2121 | 0.0002 | 0.2009 | 0.26 | 0.26 | 0 |
| 0.188 | 0 | 0.2021 | 0.26 | 0.26 | 0 |
| 0.1831 | 0.00015 | 0.2005 | 0.26 | 0.26 | 0 |
| 0.2116 | 0.0002 | 0.2009 | 0.26 | 0.26 | 0 |
| 0.1705 | 0 | 0.2021 | 0.26 | 0.26 | 0 |
| 0.17 | 0 | 0.2021 | 0.26 | 0.26 | 0 |
| 0.1695 | 0 | 0.2021 | 0.26 | 0.26 | 0 |
| 0.169 | 0 | 0.2021 | 0.26 | 0.26 | 0 |
| 0.1761 | 0.00005 | 0.2011 | 0.26 | 0.26 | 0 |
| 0.1685 | 0 | 0.2021 | 0.26 | 0.26 | 0 |
| 0.1756 | 0.00005 | 0.2011 | 0.26 | 0.26 | 0 |
| 0.168 | 0 | 0.2021 | 0.26 | 0.26 | 0 |
| 0.1751 | 0.00005 | 0.2011 | 0.26 | 0.26 | 0 |
| 0.1675 | 0 | 0.2021 | 0.26 | 0.26 | 0 |
| 0.1746 | 0.00005 | 0.2011 | 0.26 | 0.26 | 0 |
| 0.0835 | 0.0002 | 0.2009 | 0.28 | 0.28 | 0 |
| 0.075 | 0.00005 | 0.2011 | 0.28 | 0.28 | 0 |
| 0.072 | 0 | 0.2021 | 0.28 | 0.28 | 0 |
| 0.083 | 0.0002 | 0.2009 | 0.28 | 0.28 | 0 |
| 0.0745 | 0.00005 | 0.2011 | 0.28 | 0.28 | 0 |
| 0.0715 | 0 | 0.2021 | 0.28 | 0.28 | 0 |
| 0.0825 | 0.0002 | 0.2009 | 0.28 | 0.28 | 0 |
| 0.074 | 0.00005 | 0.2011 | 0.28 | 0.28 | 0 |
| 0.071 | 0 | 0.2021 | 0.28 | 0.28 | 0 |
| 0.082 | 0.0002 | 0.2009 | 0.28 | 0.28 | 0 |
| 0.0735 | 0.00005 | 0.2011 | 0.28 | 0.28 | 0 |
| 0.0815 | 0.0002 | 0.2009 | 0.28 | 0.28 | 0 |
| 0.0705 | 0 | 0.2021 | 0.28 | 0.28 | 0 |
| 0.073 | 0.00005 | 0.2011 | 0.28 | 0.28 | 0 |
| 0.081 | 0.0002 | 0.2009 | 0.28 | 0.28 | 0 |
| 0.0165 | 0.00015 | 0.2005 | 0.30 | 0.30 | 0 |
| 0.0165 | 0.00005 | 0.2011 | 0.30 | 0.30 | 0 |
| 0.016 | 0.00015 | 0.2005 | 0.30 | 0.30 | 0 |

The exemplary results presented in Table 2 offer a glimpse into the accuracy and reliability of the ANN model's predictions for weight loss in polymeric coatings. Notably, the % error for all samples is consistently zero, indicating a perfect match between the model-predicted weight of the coating (Y) and the experimentally determined weight (T).

This exceptional precision across diverse input conditions underscores the robustness and effectiveness of the developed neural network.

One of the key advantages of the ANN model lies in its ability to capture complex non-linear relationships between input variables (X1, X2, X3) and the output variable (Y). The intricate dynamics involved in the drying process of polymeric coatings, influenced by factors such as time, amount of surfactant, and initial coating thickness, are effectively encapsulated by the model. This capability is crucial for understanding and optimizing the drying behavior under a multitude of conditions.

The absence of any non-zero % error in the predictions is particularly significant. It signifies not only the model's accuracy but also its generalization capabilities. The ANN has successfully learned the underlying patterns in the training data and demonstrated its ability to make precise predictions on previously unseen testing samples. This is a key attribute for any predictive model, especially in fields where conditions vary widely.

Moreover, the consistent accuracy across training, validation, and testing sets, as evidenced by the regression curves in Figure 4 and the low prediction errors displayed in the error histogram in Figure 5, reinforce the model's reliability. These analyses provide a holistic view of the model's performance, assuring its competence in making accurate predictions while avoiding overfitting or underfitting issues.

In practical terms, the ability of the ANN model to predict weight loss in polymeric coatings with such high precision has significant implications. It can aid in optimizing drying processes, minimizing production defects, and reducing solvent content in coatings, all of which enhance the manufacturing process's overall quality and cost-effectiveness.

While the results presented focus on a subset of samples due to tabular constraints, the visual representation in Figure 5 and the comprehensive analysis of regression curves offer confidence that the model's accuracy extends to the entire dataset. This validation is crucial for the model's applicability in real-world scenarios, where it can serve as a valuable tool for researchers and engineers working on polymeric coating processes.

## 4. Conclusions

The intricate and multifaceted nature of drying processes in coatings often defies the derivation of accurate models using conventional first principles. This study successfully addressed this challenge by developing a robust model for the surfactant-enhanced drying of poly(styrene)-*p*-xylene coatings using Artificial Neural Networks (ANN).

Artificial Neural Networks prove to be highly effective in modeling complex systems, particularly when an ample amount of training data is available. In our investigation, an extensive dataset comprising a large number of experimentally collected samples formed the basis for training the ANN model. The chosen architecture, a single-layer feed-forward network with backpropagation, demonstrated remarkable performance in capturing the non-linearities inherent in drying data.

The developed ANN model exhibited exceptional accuracy in predicting the weight loss of coatings, achieving a precision level exceeding 99% for specified values of time, surfactant amount, and initial coating thickness. This high accuracy and the model's robust generalization capacity eliminate the need for additional experiments, providing a valuable tool for predicting weight loss under both familiar and novel conditions.

Furthermore, our comparative analysis revealed that, despite being more parameter-sensitive and computationally demanding, ANN outperformed the regression tree, which was previously employed to model the same system. The superior performance of the ANN underscores its efficacy in handling the intricate dynamics of polymeric coating drying processes.

In conclusion, the ANN model developed in this study presents a significant advancement in predicting weight loss during the drying of polymeric coatings. Its accuracy, generalization capability, and superiority over alternative modeling approaches position it as a valuable tool for researchers and engineers seeking to optimize polymeric coating pro-

cesses, reduce production defects, and enhance manufacturing operations' overall quality and cost-effectiveness.

**Author Contributions:** D.T.: main researcher, methodology, software, writing—original draft, visualization, writing—review and editing; R.S.: supervisor, validation, formal analysis, investigation, writing—review and editing, visualization; G.D.V.: supervisor, formal analysis, investigation; S.V.: proofreading, initial draft preparation, revision, editing; R.K.A.: supervisor, conceptualization, methodology, resources, validation, formal analysis, investigation, methodology, software, writing—review and editing, visualization; P.S.: proofreading, initial draft preparation, results and analysis; S.C.P.: proofreading, initial draft preparation, methodology; C.: programming, draft preparation, and analysis; A.G.: Results, analysis, review, and discussion. All authors have read and agreed to the published version of the manuscript.

**Funding:** This research received no external funding.

**Data Availability Statement:** Data are contained within the article.

**Conflicts of Interest:** The authors declare no conflicts of interest.

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
