# Peer review of "Modeling of Triphenyl Phosphate Surfactant Enhanced Drying of Polystyrene/p-Xylene Coatings Using Artificial Neural Network"

_processes, doi:10.3390/pr12020260_

Round 1

Reviewer 1 Report

Comments and Suggestions for Authors

1.      The main findings should be added in the abstract.

2.      The gap and novelty of the work should be mentioned.

3.      It is recommended to mention and review the other modeling methods such as RSM in the petroleum industry. For this purpose, the next work can be used in the revision stage: https://doi.org/10.1007/s13202-023-01679-2

4.      In Figure 3, R=1, it is not correct. It should be written R2=1.

5.      The quality of figures 2 and 3 should be improved.

6.      It is better to present the training and test data values used in ANN.

7.      The best validation performance graph can improve the quality of the work. If it is possible, please add it.

8.      The histogram graph of errors for training, test and validation is also recommended to use.

9.      The conclusion is written incorrectly. Please present in this section what you found in this work. The present form is not acceptable

Author Response

Responses are attached.

Reviewer 2 Report

Comments and Suggestions for Authors

The submitted article covers the application of ANN to model surfactant-enhanced drying in binary solution. The empirical data, sourced from the authors' earlier research, should be systematically incorporated into a supplementary table for enhanced clarity. There are too many self-citations, therefore, and I suggest adding more references.

Specific comments:

Page 1, line 14, provide background information in the abstract.

Page 3, line 123, highlight input and output variables for the ANN model.

Pages 4, 5 and 6, necessitate an expansion of references to fortify the academic underpinning of the work, provide additional references.

Pages 6-9, In the results and discussion section there is no discussion, just results.

Consider performing optimization using ANN.

In the references list, a refinement should be made: correct year to bold and volume to italics.

Author Response

Responses are attached.

Reviewer 3 Report

Comments and Suggestions for Authors

The authors have applied robust and versatile artificial neural network to model complex drying of surfactant enhanced drying of polymeric system. The authors have used triphenyl phosphate to enhance the drying rates in order to delay the glassy layer formation for maximum solvent recovery. The authors have demonstrated model predictions with the excellent agreement of the experimental data. Overall, this work can inspire more research design ideas of polymer. Therefore, I would like to recommend this work to publish in Processes. Below are some comments for the authors.

1. For sections of “2. Machine Learning Techniques” and “3. Model Development”, these two sections would be better to arrange as “Materials and Methods”.

2. For Figure 3, the authors have mentioned Data, Fit, and Y =T (inset). However, the figures have no such information. Please correct Figure 3.

Author Response

Responses are attached.

Reviewer 4 Report

Comments and Suggestions for Authors

This paper deals with triphenyl phosphate surfactant enhanced drying of polystyrene/p-xylene coating via an artificial neural network modeling. It is recommended to accept this paper for publication after some revision on the basis of comments below.

COMMENTS

1.

The title must be more specific. The following title is recommended:

Modeling of Triphenyl Phosphate Surfactant Enhanced Drying of Polystyrene/p-Xylene Coatings using Artificial Neural Network

2.

Polystyrene is one word according to IUPAC nomenclature rules, that is, parenthesis is not required. The authors should correct this in the whole manuscript.

3.

In line 269, the authors write that “All 16,258 samples (obtained experimentally) were loaded to Neural Net Fitting software”. However, they do not provide the sources (references) of these data. This must be provided.

4.

The authors do not provide any information on what kind of data, that is, physically what kind of measured parameters, were applied for this model. It is very important to provide detailed information on the applied parameters (data) in this study.

5.

In lines 271-272, the authors write that “we selected 70% (11380) samples for training, 15% (2439) for validation and 15% (2439) for testing”. However, no any explanation is provided why this ratio of data selection was made. It is known that the ratio of the training/validation/testing data has influence on the outcome generated by this method. The authors have to explain why this weighing of the data was selected by them.

6.

In line 313, correctly straight   and not   strait.

7.

In Table 2, presenting the data with more than three valuable digits is a nonsense (Y is even shown by 9 digits). This should be corrected by the authors.

8.

The authors do not describe and explain the meaning of the parameters, such as X1, X2, TPP, T and Y,  in Table 2. These should be precisely described and explained in details in the text.

9.

The authors claim in the Abstract that “The model predictions can be used to predict results for unknown compositions with great degree of confidence and will be helpful for the researchers working in the field of polymer.” However, the polymer-solvent, polymer-surfactant and solvent-surfactant interactions depends on the chemical nature (structure) of these compounds, not to mention the applied concentrations as the authors also mention, and therefore, it is unclear how the applied model can be used without taking into account the individual strength of these interactions which are determined by the chemical structure of the components. The authors should discuss and explain this in details.

Author Response

Responses are attached.

Reviewer 5 Report

Comments and Suggestions for Authors

1.  It is very interesting, how it was proved that the functional minima obtained are unique, but not local?

2.  Are the plots in Figure 3 curves? After altering any scale or both at once, the angles (450) of these lines will change as well.

3.  Table 2. The number of significant digits in T and Y is incorrect.

4.  I still haven’t figured out what the novelty of the work is. It seems that the novelty of the research is only the application of ANN approach to drying binary solutions. Isn’t it? If this is not the case, dear authors are encouraged to explain more clearly the purpose of their work and what is the improvement of their research compared to [6-9] and similar studies by other authors on the matter.

Comments on the Quality of English Language

English is very specific but understandable

Author Response

Responses are attached.

Round 2

Reviewer 1 Report

Comments and Suggestions for Authors

Thanks for responding to the comments. The work was improved. Regarding to comment 3, the other modeling techniques such as RSM was not mentioned and the recommended reference (https://doi.org/10.1007/s13202-023-01679-2) was not used. Please mention it  and use this reference.

Author Response

Thanks for responding to the comments. The work was improved. Regarding to comment 3, the other modeling techniques such as RSM was not mentioned and the recommended reference (https://doi.org/10.1007/s13202-023-01679-2) was not used. Please mention it  and use this reference.

Response

Thanks for your suggestion. Your suggestion and reference has been added.

Before the emergence of machine learning techniques, researchers have been using design of experiments (DOE) methods for determining the relationship between various factors affecting the process and its outputs in other terms, causal analysis is used to identify cause-and-effect relationships. There are multiple approaches to DOE, like Taguchi and response surface methodology (RSM). For example, the precipitation of barium sulfate salt was investigated through dynamic tube blocking tests using RSM [29].

   After the emergence of artificial intelligence and machine learning techniques (MLTs), more competitive methods have been developed which can be adopted to compliment or replace traditional RSM methods. 

Reviewer 2 Report

Comments and Suggestions for Authors

The author's responses are fair. The article is improved. 

Author Response

The author's responses are fair. The article is improved. 

Response

Thanks for your appreciation and acceptance of our work.